# TARGETED SYNTHETIC DATA GENERATION FOR TABULAR DATA VIA HARDNESS CHARACTERIZATION

## ABSTRACT

Synthetic data generation has been proven successful in improving model performance and robustness in the context of scarce or low-quality data. Using the data valuation framework to statistically identify beneficial and detrimental observations, we introduce a novel augmentation pipeline that generates only high-value training points based on hardness characterization. We first demonstrate via benchmarks on real data that Shapley-based data valuation methods perform comparably with learning-based methods in hardness characterisation tasks, while offering significant theoretical and computational advantages. Then, we show that synthetic data generators trained on the hardest points outperform non-targeted data augmentation on simulated data and on a large scale credit default prediction task. In particular, our approach improves the quality of out-of-sample predictions and it is computationally more efficient compared to non-targeted methods.

## 1 INTRODUCTION

Training complex machine learning models requires large amounts of data, but in real-world applications data may be of poor quality, insufficient in amount, or subject to privacy, safety, and regulatory limitations. Such challenges have sparked an interest in *synthetic data generation* (SDG), representing the practice of using available data to generate realistic synthetic samples (Lu et al., 2024). In this work, we argue that, when the objective is to use synthetic data to make an existing machine learning model better generalize to unseen data, augmenting only the hardest training points is more effective than augmenting the entire training dataset. In other words, we propose a *scalable targeted synthetic data generation* framework, focusing on binary classification tasks. The underlying intuition is that, within a dataset, some of the observations are obvious to classify, whereas others play a more significant role in determining the decision boundary of the trained model. Focusing only on these harder examples when generating additional data may allow the model to efficiently learn a more robust decision boundary.

In particular, the method proposed in this work is specifically targeted towards binary classifiers on tabular data, which often require adaptations such as oversampling, undersampling, or synthetic data generation to improve model performance (Chawla et al., 2002; He et al., 2008) under class imbalance. While deep neural networks have proven successful across domains such as images, audio or text, they are still regularly outperformed on tabular data by simpler and more interpretable tree-based architectures (Shwartz-Ziv & Armon, 2021). Therefore, we choose trees as the reference model for binary classification on tabular data, while deep learning models are used for synthetic data generation. In general, we will use $D = \{(\boldsymbol{x}_i, y_i)\}_{i=1}^n$ to denote the training data, where $\boldsymbol{x}_i \in \mathbb{R}^d$ is a $d$-dimensional feature vector and $y_i \in \{0, 1\}$ is a binary label.

Our targeted synthetic data generation approach is divided into two main steps: *(i)* hardness characterization of training data points, followed by *(ii)* training of a synthetic data generator only on the hardest data points of the initial training dataset. In order to identify "hard" data points, we propose a novel hardness characterizer specific to tabular data, based on KNN data Shapleys (Jia et al., 2020), capable of achieving performances comparable to state-of-the-art methods in hardness detection benchmarks, while offering key advantages such as being deterministic, model agnostic and computationally efficient. Hardness characterization has already been applied in the literature with rationales similar to this work, but mainly with the objective of pruning the easier examples (Jia et al., 2021; Paul et al., 2023), rather than augmenting the harder ones.

Once the hardest points have been identified, synthetic data generation models are trained only on the most difficult points identified by our hardness characterizer. In this work, we focus on two widely used deep learning architectures for synthetic data generation for tabular data, namely Tabular Variational Autoencoders (TVAE) and Conditional Tabular Generative Adversarial Networks (CTGAN) (both introduced in Xu et al., 2019). To mitigate instabilities encountered during training, we devise a custom early stopping condition which allows to interrupt training when the quality of the synthetically generated samples is maximised. The best synthesizers are then found using a highly efficient routine which combines model selection and hyperparameter tuning. From a comparison of the best models we find that, given a fixed number of points to generate, augmenting only hard data results in a larger performance improvement on unseen data with respect to non-targeted data augmentation, while being computationally cheaper.

The rest of the work is structured as follows: after a recap on the related literature in Section 2, the proposed approach is discussed and empirically verified in Section 3. Next we explore the use of KNN Shapleys as hardness characterizers via a benchmark study in Section 4.1, followed by a discussion on how to utilize them for synthetic data generation on real (Section 4.2) and simulated data (Section 4.3). The results are extensively discussed in Section 5.

## 2 BACKGROUND AND RELATED LITERATURE

In this work, we aim to bridge the gap between existing methods for hardness characterization and game theoretic approaches to data valuation, with the objective of improving the performance of synthetic data generators. Therefore, our work closely aligns with existing techniques in hardness characterization, data valuation, and synthetic data generation, briefly reviewed in this section.

### 2.1 HARDNESS CHARACTERIZATION

As remarked in the introduction, the first aim of this work is the identification of a subset of points that are hard for the model to learn. Existing literature on the topic of hardness characterization is often qualitative and based on different definitions of "hardness". Following the taxonomy presented in Seedat et al. (2024), three main categories of hardness can be identified: *(i) Mislabeling*: the true label is replaced with an incorrect one. The label perturbation probability can be uniform across classes or label-dependent; *(ii) Out-of-distribution*: features are transformed or shifted, leading to observations distributed differently from the main data generating process; *(iii) Atypical*: observations that are compatible with the main data generating process, but located in tails or rare regions of the feature distributions. These distinctions normally dictate different courses of actions: datapoints that are mislabeled or out of distribution are symptomatic of errors in the data collection process and should be removed from training data, while atypical datapoints are valid points and may in fact need to be augmented to train a robust model (Yuksekgonul et al., 2023).

While many different hardness characterizers have been proposed, most of them are based around keeping track of some statistic of interest for each datapoint during the training process, meaning they can be adapted to any model that is trained iteratively (such as XGBoost or a neural network). Examples include *GraNd* (Paul et al., 2023), which uses large gradients as an indicator of hardness, and *Forget* (Toneva et al., 2019), which counts how many times each point is learnt and then forgotten during training, with the simple intuition that hard points are forgotten more often. Based on the quantitative benchmarks reported in Seedat et al. (2024), *Data-IQ* (Seedat et al., 2022) emerges as one of the best performers on tabular data, and it is chosen as benchmark hardness characterizer in this work because of its interpretability and compatibility with XGBoost.

**Data-IQ** Data-IQ (Seedat et al., 2022) partitions the training data into "easy" and "hard" points via the concepts of *epistemic* and *aleatoric* uncertainty, combined with a model confidence score calculated from $\mathcal{P}(\boldsymbol{x}, \boldsymbol{\theta})$, representing the probability of assigning a correct label to an observation, given input features $\boldsymbol{x}$ and model parameters $\boldsymbol{\theta}$. The epistemic uncertainty is caused by the variability in the estimated model parameters, whereas the aleatoric uncertainty captures the inherent data uncertainty. Given parameter estimates $\{\boldsymbol{\theta}_e\}_{e=1}^{E}$ obtained during training, a Data-IQ *confidence score* is obtained as $\bar{\mathcal{P}}(\boldsymbol{x}) = E^{-1} \sum_{e=1}^{E} \mathcal{P}(\boldsymbol{x}, \boldsymbol{\theta}_e)$, whereas the aleatoric uncertainty is given by $v_{\mathrm{al}}(\boldsymbol{x}) = E^{-1} \sum_{e=1}^{E} \mathcal{P}(\boldsymbol{x}, \boldsymbol{\theta}_e)[1 - \mathcal{P}(\boldsymbol{x}, \boldsymbol{\theta}_e)]$. Training data can then be partitioned into *(i) easy*

points, corresponding to high $\bar{\mathcal{P}}(\boldsymbol{x})$ and low $v_{\text{al}}(\boldsymbol{x})$, *(ii) hard* points, corresponding to low $\bar{\mathcal{P}}(\boldsymbol{x})$ and low $v_{\text{al}}(\boldsymbol{x})$, and *(iii) ambiguous* points, with high $v_{\text{al}}(\boldsymbol{x})$.

## 2.2 DATA VALUATION

Data valuation (see Ghorbani & Zou, 2019) is the task of equitably quantifying the contribution of each data point to a model, via a data value $\phi_i(D, \mathcal{A}, V)$, where $D$ is the training dataset, $\mathcal{A}$ is a model and $V$ is a performance score, where $V(S)$ is the performance score of $\mathcal{A}$ when trained on $S \subseteq D$. Ghorbani & Zou (2019) identifies that a suitable data valuator is the *data Shapley* of the $i$-th training point, which takes the following form:

$$\phi_i = C \sum_{S \subseteq D \setminus \{i\}} \binom{n-1}{|S|}^{-1} [V(S \cup \{i\}) - V(S)], \tag{1}$$

where $C$ is an arbitrary constant. The valuator $\phi_i$ can be interpreted as weighted sum of all the possible "marginal contributions" of the datapoint. Exact evaluation of the data Shapleys is prohibitively expensive because of the need to retrain the model $2^{n-1}$ times (once per subset) for each observation in the training data. It is possible to approximate the data Shapleys by considering only a small number of subsets $S \subseteq D \setminus \{i\}$ in equation 1, obtained via Monte-Carlo sampling of random permutations. Unfortunately, even state-of-the-art approximations based on this idea, such as *TMC-Shapley* (Ghorbani & Zou, 2019) or *Beta-Shapley* (Kwon & Zou, 2022) are computationally prohibitive for datasets with $n \gg 1000$, making them difficult to use in practice. More generally, extensive work has been developed in the literature to make the computation of data Shapleys more efficient, such as Jia et al. (2019), Wang et al. (2023) and Wang et al. (2024).

In particular, Jia et al. (2020) derived a recursive algorithm to compute the exact data Shapleys for KNN classifiers in $\mathcal{O}\{n \log(n) \, n_{\text{test}}\}$ complexity. For training data $\{(\boldsymbol{x}_i, y_i)\}_{i=1}^{n}$ and test data $\{(\boldsymbol{x}_{\text{test},i}, y_{\text{test},i})\}_{i=1}^{n_{\text{test}}}$, let $(\alpha_{j,1}, \ldots, \alpha_{j,n})$ be the indices of training data points, sorted in increasing order according to their Euclidean distance from the $j$-th test data point, for $j = 1, \ldots, n_{\text{test}}$. The KNN Shapley for the $i$-th training data point is then obtained as the average $s_i = n_{\text{test}}^{-1} \sum_{j=1}^{n_{\text{test}}} s_{j,i}$, where $s_{j,i}$ is calculated recursively as follows:

$$s_{j,\alpha_{j,n}} = \frac{1}{n} \mathbb{1}_{y_{\text{test},j}} \{y_{\alpha_{j,n}}\}, \qquad s_{j,\alpha_{j,i}} = s_{j,\alpha_{j,i+1}} + \frac{\mathbb{1}_{y_{\text{test},j}} \{y_{\alpha_{j,i}}\} - \mathbb{1}_{y_{\text{test},j}} \{y_{\alpha_{j,i+1}}\}}{iK \cdot \min\{K, i\}^{-1}}, \tag{2}$$

where $j = 1, \ldots, n_{\text{test}}$, $i = 1, \ldots, n-1$, and $\mathbb{1}.\{\cdot\}$ is the indicator function. Jia et al. (2021) found KNN Shapleys to be a valid alternative to data Shapleys in tasks such as data summarization or noisy labels detection, only when based on deep features extracted from image data by pre-trained architectures. A similar approach is unfortunately not possible on tabular data. In this work we argue that KNN Shapleys computed using a test dataset large enough to cover the entire data distribution, can be used for hardness characterization on tabular data. Specifically, we argue that the *lowest* KNN Shapleys identify the *hardest* training points.

## 2.3 SYNTHETIC DATA GENERATION

Synthetic data generators (SGDs) for tabular data have been actively researched in recent years (see for example Fonseca & Bacao, 2023). Here, we discuss three of the most popular SDGs used in practice: SMOTE, CTGAN, and TVAE.

**SMOTE** In *Synthetic Minority Over-sampling Technique* (SMOTE; Chawla et al., 2002), observations are picked at random among the $K$ nearest neighbours of the same class for each datapoint, and new data is generated by random sampling along the segment connecting the point to the chosen neighbour. The number of neighbours is determined based on the over-sampling ratio.

**CTGAN** *Tabular GAN* (TGAN; Xu & Veeramachaneni, 2018) was the first attempt at using *Generative Adversarial Networks* (GAN; Goodfellow et al., 2014) to generate tabular data, which was then extended to *Conditional TGANs* (CTGANs; Xu et al., 2019). A GAN consists of two models: a generator that takes as input random noise and tries to output a point following the data distribution, and a discriminator that given observations (either from the original dataset or synthesized

by the generator) outputs the probability of them being fake. Both the generator and discriminator are typically neural networks. In CTGANs, the possible non-Gaussianity and multimodality in the data is handled by fitting a variational Gaussian mixture model to each normalized numeric feature, whereas categorical features are represented as one-hot encoded vectors. Both the generator and the discriminator networks within the underlying GAN are multilayer perceptrons (MLPs). The name of the CTGAN architecture derives from the form of its conditional generator, which allows to generate data conditional on specific values of the discrete features, for better handling imbalanced datasets.

**TVAE** A *Tabular VAE* (TVAE; Xu et al., 2019) is an adaptation for tabular data of a Variational Auto-Encoder (VAE; Kingma & Welling, 2013), using similar preprocessing to CTGAN. A VAE consists in an encoder that learns to represent high-dimensional data into a low-dimensional latent space, and a decoder that reconstructs the compressed representation into the original domain. Both the encoder and decoder networks are normally chosen to be MLPs (Kingma & Welling, 2013).

## 3 KNN SHAPLEYS AS HARDNESS CHARACTERIZERS FOR SDGS

In this work, we propose to utilize KNN Shapleys as hardness characterizers, and subsequently generate synthetic data by only augmenting the hardest points. Our proposed pipeline is the following:

1. **KNN Shapley calculation** – For a model $\mathcal{A}$ fitted on training data $D = \{(\boldsymbol{x}_i, y_i)\}_{i=1}^n$, calculate the KNN Shapleys $s_i,\ i = 1, \ldots, n$, based on a test set $\{(\boldsymbol{x}_{\text{test},i}, y_{\text{test},i})\}_{i=1}^{n_{\text{test}}}$.

2. **Ranking by hardness** – Sort the KNN Shapleys in *increasing order*, with points with lower values representing harder examples.

3. **Targeted augmentation** – Given a synthetic data generator $\mathcal{G}$, perform data augmentation only on the hardest points, based on the ranking of the KNN Shapleys.

By generating synthetic data for difficult examples only, we aim to improve the performance of the model $\mathcal{A}$ specifically on the most challenging parts of the data distribution. The proposed procedure is summarized visually in Figure 1. In the next section, we provide a mathematical intuition on the interpretation of KNN Shapley as hardness charaterizers.

### 3.1 KNN SHAPLEYS FOR HARDNESS CHARACTERIZATION: A TOY EXAMPLE

Consider a mixture of two univariate normals with unit variance, centred at $-1$ and $1$, such that $p(x \mid y = 0) = \mathcal{N}(x; -1, 1)$ and $p(x \mid y = 1) = \mathcal{N}(x; 1, 1)$. Datapoints are drawn from each distribution with probability $1/2$. Additionally, consider two points $(x_1, y_1) = (-1, 0)$ and $(x_2, y_2) = (1, 1)$, representative of each component of the mixture distribution. Consider a third point $(x_{\text{train}}, y_{\text{train}})$, with $y_{\text{train}} = 0$ without loss of generality. Hence, the training set takes the form $D = \{(-1, 0), (1, 1), (x_{\text{train}}, 0)\}$. As $x_{\text{train}}$ increases, it becomes *harder* to classify correctly under the true underlying data distribution, since the ratio $p(x_{\text{train}} \mid y = 0) \ / \ p(x_{\text{train}} \mid y = 1)$ decreases, consequently increasing the probability of misclassification. Note that any value of $x_{\text{train}} \in \mathbb{R}$, combined with $x_1$ and $x_2$, partitions the real line into four different regions.

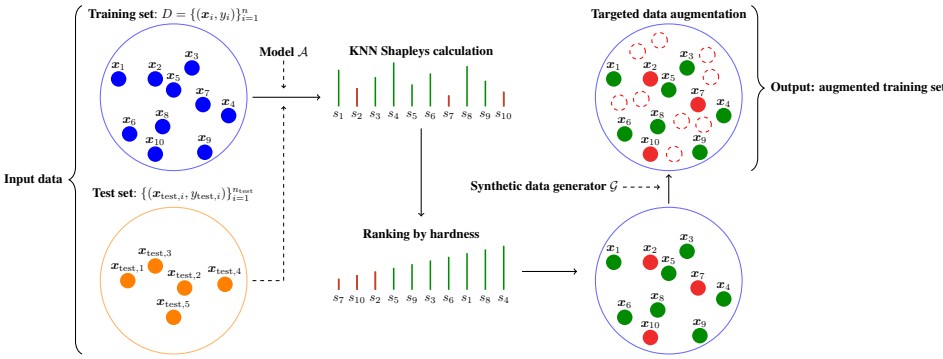

Figure 1: Visual representation of the proposed targeted synthetic data generation pipeline.

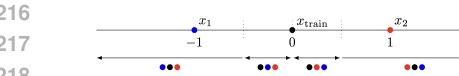

(a) Four regions for $x_{\text{test}}$ and ranking of $\{x_1, x_{\text{train}}, x_2\}$, for $x_{\text{train}} = 0$.

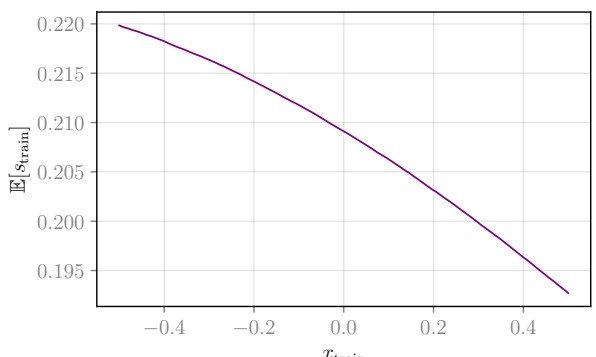

| $x_{\text{test}}$ | $y_{\text{test}}$ | $s_{-1}$ | $s_{\text{train}}$ | $s_1$ |
|---|---|---|---|---|
| $(-\infty, -1.2)$ | 0 | 1/2 | 1/2 | 0 |
| $(-\infty, -1/2)$ | 1 | -1/6 | -1/6 | 1/3 |
| $(-1/2, 0)$ | 0 | 1/2 | 1/2 | 0 |
| $(-1/2, 0)$ | 1 | -1/6 | -1/6 | 1/3 |
| $(0, 1/2)$ | 0 | 1/3 | 5/6 | -1/6 |
| $(0, 1/2)$ | 1 | 0 | -1/2 | 1/2 |
| $(1/2, +\infty)$ | 0 | 1/3 | 1/3 | -2/3 |
| $(1/2, +\infty)$ | 1 | 0 | 0 | 1 |

(b) 1NN Shapleys on the training data for $(x_{\text{test}}, y_{\text{test}}) \in \mathbb{R} \times \{0, 1\}$, $x_{\text{train}} = 0$.

(c) Expected value $\mathbb{E}(s_{\text{train}})$ of the 1NN Shapley for the data point $(x_{\text{train}}, 0)$, for different values of $x_{\text{train}}$.

Figure 2: Results for the toy experiment in Section 3.1, based on a 50-50 mixture of two univariate normals with unit variance and means $-1$ and $1$, and training data $\{(-1, 0), (1, 1), (x_{\text{train}}, 0)\}$.

Any test data point $(x_{\text{test}}, y_{\text{test}}) \in \mathbb{R} \times \{0, 1\}$ induces a ranking of the training data based on the distances $|x_{\text{test}} - x|$, $x \in \{x_1, x_2, x_{\text{train}}\}$. For $x_{\text{train}} = 0$, the four possible regions with different rankings of the three training points are plotted in Figure 2a. The 1NN Shapley value $s_{\text{train}}$ for the point $(x_{\text{train}}, y_{\text{train}}) = (0, 0)$ can be calculated explicitly for all the possible values of $(x_{\text{test}}, y_{\text{test}})$. The results are reported in Table 2b. Since the data distribution is known, the expected value $\mathbb{E}(s_{\text{train}})$ of the 1NN Shapley for $(x_{\text{train}}, y_{\text{train}}) = (0, 0)$ can be explicitly calculated:

$$\mathbb{E}(s_{\text{train}}) = \int_{-\infty}^{+\infty} s_{\text{train}}\, p(x_{\text{test}})\, \mathrm{d}x_{\text{test}} = \frac{1}{2} \sum_{y \in \{0,1\}} \int_{-\infty}^{+\infty} s_{\text{train}}\, p(x_{\text{test}} \,|\, y)\, \mathrm{d}x_{\text{test}} \approx 0.209. \quad (3)$$

It must be remarked that the value of $s_{\text{train}}$ depends on $(x_{\text{test}}, y_{\text{test}})$, as detailed in Table 2b and equation 1. The same procedure as equation 3 can be repeated for any $x_{\text{train}} \in \mathbb{R}$, with results displayed in Figure 2c. As expected, the 1NN Shapley decreases in the direction of increasing hardness, suggesting an association between low KNN Shapleys and hard regions of feature space.

## 4 RESULTS

In this section, the proposed hardness characterizer based on KNN Shapleys is first compared to the most common methods in the literature via comprehensive benchmarks on tabular data. Then, KNN Shapleys are calculated on a large credit default prediction dataset to identify the hardest points, which are later used to train and compare synthetic data generators. Lastly, the same pipeline is applied to a simulated dataset to verify consistency in the results.

### 4.1 BENCHMARK STUDY

In Seedat et al. (2024), the problem of hardness characterization is approached quantitatively by comparing existing methods on how confidently they can identify different kinds of hard datapoints on a variety of OpenML datasets (Vanschoren et al., 2013). Their toolkit supports two tabular datasets (*diabetes* and *cover*) and it allows to perturb a chosen proportion $p$ of datapoints according to one of three hardness types (*mislabeling*, *out-of-distribution*, or *atypical*; *cf.* Section 2.1). For this study, we use $p \in \{0.05, 0.1, 0.15, 0.2\}$ and the performance metric is the *Area Under the Precision Recall Curve* (AUPRC) of an MLP. Results are averaged across three independent runs per hardness type and then across the different hardness types. The KNN Shapleys-based characterizer is implemented by adding a custom class to the original GitHub repository (https://github.com/seedatnabeel/H-CAT). The results are plotted in Figure 3.

On the diabetes dataset, our characterizer outperforms existing methods for all choices of $p$, although based on the critical difference diagram in Figure 3a, it can be noticed that performance is not significantly different from Data-IQ, the other method to offer native XGBoost support. On the

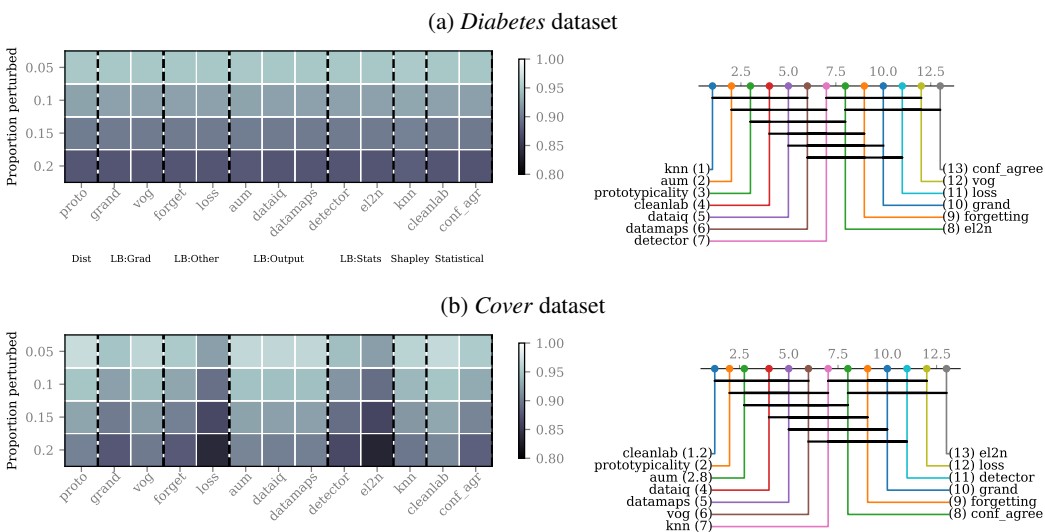

Figure 3: AUPRC heatmap and critical difference diagram for 13 different hardness characterizers (including KNN Shapleys, labelled `knn`) on the *diabetes* and *cover* datasets (Seedat et al., 2024).

cover dataset (see Figure 3b), our characterizer ranks 7th overall, although once again performance is statistically compatible with Data-IQ. Based on the mathematical intuition discussed in Section 3.1, KNN Shapleys are expected to be lower when a datapoint lies in a region of feature space where its target is *atypical* (*cf.* Section 2.1). When $p$ is too large relative to the dataset's difficulty, this atypicality is compromised, causing a deterioration in the performance of our characterizer.

Based on these results, low KNN Shapleys prove to be solid indentifiers of hardness. Most hardness characterization techniques in the literature were devised for neural networks, and do not natively support XGBoost. On the other hand, KNN Shapleys have the clear advantage over existing methods of not requiring model training and thus being deterministic and model agnostic. However, they require an external test set and can become expensive to calculate on very large datasets ($n > 10^6$).

## 4.2 AMERICAN EXPRESS DATASET

The main real-world benchmark for the methodologies discussed in this work is the American Express credit default prediction dataset (Addison et al., 2022), referred to as the *Amex dataset*. The dataset consists of observations for $n = 458\,913$ customers. For each of them, we consider 188 features available at the latest credit card statement date, in addition to a binary label $y_i \in \{0, 1\}$ which specifies whether or not the customer was able to repay their credit card debt within 120 days of the statement. The $n = 458\,913$ available observations are split into $n_{\text{valid}} = 50\,000$ datapoints used to validate model performance on unseen data, $n_{\text{test}} = 50\,000$ observations used to calculate KNN Shapleys, and the remaining datapoints are used for training. The splits are performed randomly stratifying on the target, so that the same ratio of defaulters and non-defaulters is maintained across datasets. Performance is measured via the normalized Gini coefficient, which is $2 \cdot \text{AUCROC} - 1$ with AUCROC denoting the *Area Under the Receiver Operating Characteristic* curve.

The Amex dataset has been first released as part of a Kaggle[1] competition, with winning submissions typically consisting in ensembles of different models. XGBoost was found to have the best performance *on its own* and is thus chosen as baseline for this study. In order to benefit from GPU acceleration, training was carried out on a cluster equipped with 4 GeForce RTX 3090 Ti GPUs and 256GB of memory. The number of gradient boosting rounds was set to an arbitrarily high number, using early stopping on validation Gini with a patience of 100 iterations of gradient boosting. Because of subsampling the datapoints and the features, the training process is non-deterministic and thus retraining a large number of times with different random seeds is required to mitigate random-

---

[1] See https://www.kaggle.com/competitions/amex-default-prediction.

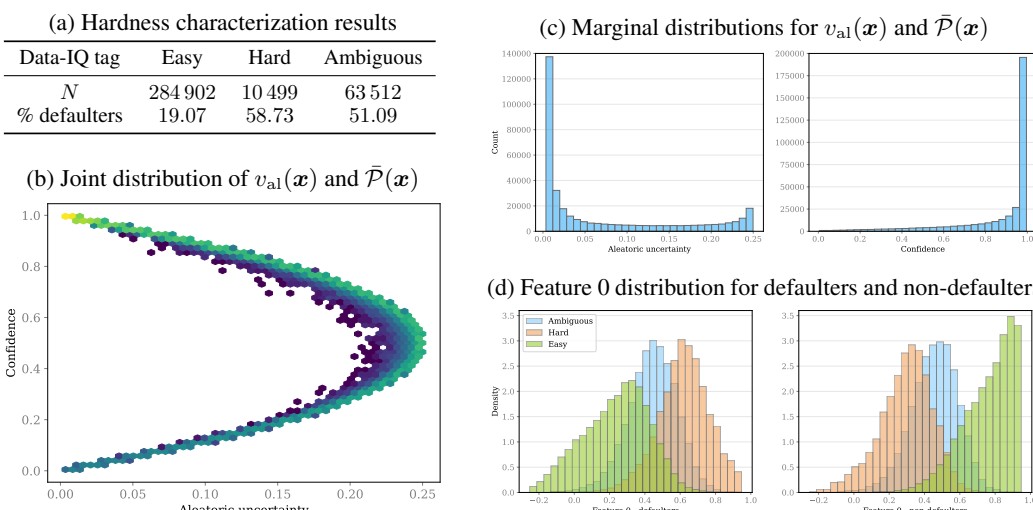

Figure 4: Data-IQ hardness characterization scores on the Amex credit default prediction dataset.

ness. This forces to choose a set of hyperparameters that offers low training times as well as solid validation performance. Through experimenting it was found, that setting the maximum tree depth to 4, the learning rate to 0.05, the subsampling ratio of datapoints to 0.8 and of features to 0.6, gives a validation Gini of $0.91986$ in under 2 minutes of training time.

### 4.2.1 HARDNESS CHARACTERIZATION

Data-IQ's confidence scores and aleatoric uncertainties are estimated for each training observation as per Section 2.1. Figure 4b displays the relationship between the two: in the top left a large amount of "easy" points, with high confidence and low aleatoric uncertainty; in the bottom left a few "hard" points, with low confidence and low aleatoric uncertainty; lastly, the "ambiguous" points are located around the elbow, with high aleatoric uncertainty. Looking at the marginal distributions in Figure 4c, it is noticed that XGBoost is generally very confident in its predictions, causing the number of hard points to be low. Setting a low confidence threshold of $0.25$, a high confidence threshold of $0.75$ and a low aleatoric uncertainty threshold of $0.2$ allows to assign to each point a tag in $\{Easy, Hard, Ambiguous\}$. Hard points are less than $3\%$ of training data and both hard and ambiguous points have a much higher proportion of defaulters than the full data (*cf.* Table 4a).

Beyond these initial statistics, it is interesting to know where these points lie in feature space. By looking at the total reduction in loss due to a feature in the nodes where said feature is used to split the data, we can calculate the feature importances of a trained XGBoost classifier. In the case of the Amex dataset, the first feature dominates these scores, with an importance comparable to the bottom 100 features combined. Figure 4d displays the distribution of this feature separated by Data-IQ tag for both defaulters and non-defaulters: hard defaulters somewhat overlap with easy non-defaulters and vice-versa, while ambiguous defaulters are almost indistinguishable from ambiguous non-defaulters. This is a common issue with tabular data, referred to as *outcome heterogeneity* and is known to be captured by Data-IQ (Seedat et al., 2022).

KNN Shapleys are then computed for $K \in \{1, 5, 100\}$ under the assumption that the $n_{\text{test}}$ available out-of-sample data points are representative of the entire data distribution. Figure 5 displays the distribution of 100NN Shapleys separated by Data-IQ's tag: hard points as per Data-IQ exhibit low 100NN Shapleys, with very good adherence for the hardest points. In addition, the left tail of the ambiguous points lies in the region of low 100NN Shapleys and would thus be recognized as hard by our novel characterizer.

To choose the best $K$, validation Gini is monitored as we re-train XGBoost after gradually removing the hardest datapoints. These points are expected to be the most valuable for XGBoost and thus the best hardness characterizer should be the one displaying the fastest drop in validation performance.

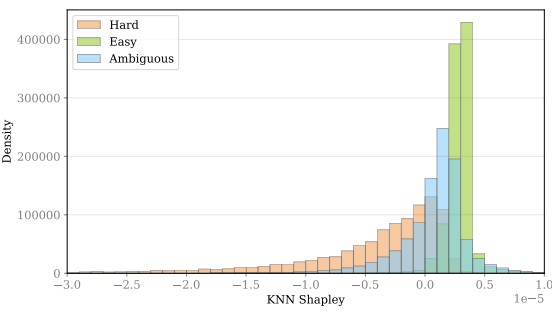
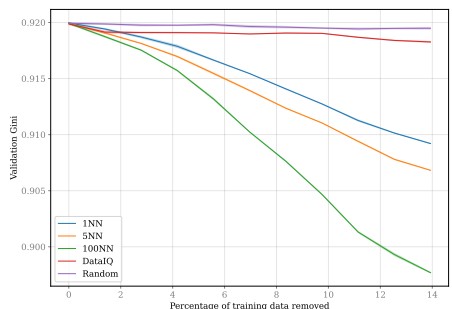

Figure 5: Distribution of the 100NN data Shapley scores for different Data-IQ tags.

Figure 6: Validation Gini after removing hardest points, with 95% CIs.

(a) CTGAN          (b) TVAE

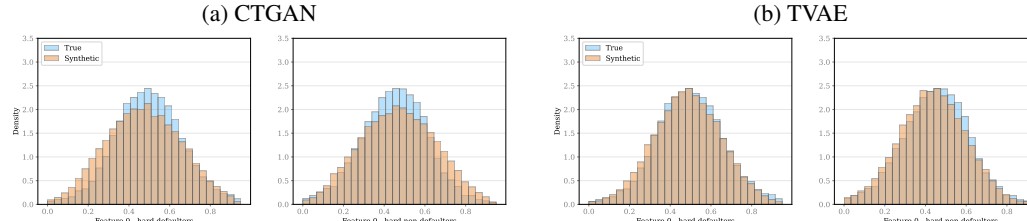

Figure 7: Histogram of feature 0 in the Amex dataset for true and synthetic data generated via CTGAN and TVAE, restricted to the 10% harest data points according to the 100NN Shapley scores.

100NN Shapleys outperform both other choices of $K$ and Data-IQ (*cf.* Figure 6). This comparison highlights an issue with Data-IQ: after a sharp drop when the hard points are removed, performance decreases at a rate comparable to random as the most ambiguous points are removed. Seedat et al. (2022) argue that the removal of ambiguous points should actually make the model more robust, as aleatoric uncertainty is data-dependent and cannot be reduced unless additional features are collected. This is however not the case for the Amex dataset, and beyond the small amount of hard points identified, Data-IQ does not provide guidance on the next most valuable points.

### 4.2.2 SYNTHETIC DATA GENERATION

The 10% hardest datapoints are chosen as training set for synthetic data generators, with the purpose of establishing whether augmenting only the hardest datapoints makes XGBoost more robust than non-targeted data augmentation. The choice of a hard threshold is imposed by the expensive computational cost of fine-tuning neural networks, and 10% is justified as it is just before the elbow region where the decrease in validation performance when removing hard data slows down (*cf.* Figure 6). Both CTGAN and TVAE are implemented in Python using a heavily customized version of the `sdv` package (Patki et al., 2016), with extensive details about the training given in Appendix A.1. The quality of the synthetic samples can be examined directly for the first feature: Figure 7a (CTGAN) displays a good but not perfect overlap with real data, whereas from Figure 7b (TVAE) we can see better overlap of synthetic and real data with respect to CTGAN for the first feature.

For models trained on the 10% hardest points we augment by 100%, while for models trained using the entire dataset we augment by 10%, thus guaranteeing the same amount of synthetic samples across all experiments. Notice that for each attempt synthetic data has to be generated multiple times with different seeds to mitigate randomness and quantify uncertainty on the performance metric. The proposed routine, implemented in Python with the `baytune` package (Smith et al., 2019), requires very few attempts to find good proposals, and is thus more efficient than a grid search over a hyperparameter space, reducing the computational complexity of the problem.

Figures 8a and 8b display the scores after 20 attempts: the best scores are in both cases achieved by TVAE, and augmenting only the hardest points generally leads to a more significant improvement

(a) Scores in fine-tuning routine: hard points

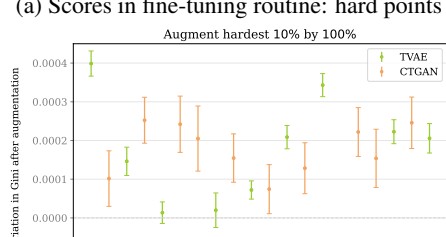

(c) Variation in Gini after augmentation with 95% CIs.

|  | hardest 10% by 100% | full data by 10% |
|---|---|---|
| SMOTE | $[-0.00122, -0.00109]$ | $[0.000116, 0.000192]$ |
| Best TVAE | $[0.000366, 0.000431]$ | $[0.000232, 0.000278]$ |
| Best CTGAN | $[0.000193, 0.000312]$ | $[0.000135, 0.000237]$ |

(d) Scores for TVAE and TVAE on hard data points after $5\%, 10\%, 15\%, 20\%$ augmentation, with $95\%$ confidence intervals estimated by generating synthetic data 10 times.

(b) Scores in fine-tuning routine: entire dataset

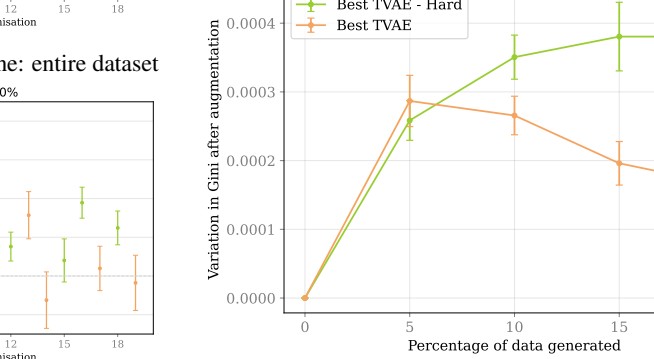

Figure 8: Gini scores under different synthetic data augmentation regimes on the Amex dataset.

in validation performance, both for the best attempts and on average across different sets of hyper-parameters. Information on the tuning grid and best hyperparameters are detailed in Appendix A.2.

Table 8c reports the performance variations for the best TVAE, CTGAN and SMOTE: the latter does not work for the hardest points and is outperformed by TVAE for general augmentation. The robustness of these results for the best attempts can be verified by augmenting by different amounts $(5\%, 10\%, 15\%, 20\%)$, with results displayed in Figure 8d: generating more hard points further increases performance with diminishing returns after $15\%$, while for non-targeted augmentation, the performance actually worsens as more data is added due to the extra noise. Finally, to quantify the magnitude of this improvement, adding back the $n_{\text{test}} = 50\,000$ observations used to compute KNN Shapleys into training data improves validation performance by $\approx 0.000326$, which is less than the improvement obtained from augmentation of the hard points via TVAE.

It could be argued that the improvement in validation performance achieved via data augmentation, although larger for models trained on only the hardest datapoints, is very small in absolute value. It must be remarked that similar improvements have only been achieved in the original Kaggle competition by generating thousands of features and building large ensemble models. Additionally, XGBoost already classifies over $90\%$ of validation data correctly, and any further improvement on this already highly performing model may have business value in practice.

## 4.3 SIMULATION STUDY

To verify the robustness of the results, we run the same hardness characterization and data augmentation pipeline on simulated data. Specifically, we consider four bivariate normal distributions, assigning to two of them label $y = 0$, and $y = 1$ to the remaining ones. We draw $n_{\text{train}} = 5\,000$ training datapoints, $n_{\text{valid}} = 2\,500$ observations for model validation and lastly $n_{\text{test}} = 2\,500$ datapoints to calculate 5NN Shapleys. The training dataset can be visualized in Figure 9a on the left, while the $5\%$ hardest points according to 5NN Shapleys are shown on the right: notice that they are concentrated around the decision boundary, with most of them falling on the "wrong" side, confirming the issue of *outcome heterogeneity* for hard data points discussed in Section 4.2.1.

We proceed by tuning TVAE using the GPEI algorithm (described in Appendix A.1) both when augmenting the hardest $5\%$ by $100\%$ and when augmenting the entire dataset by $5\%$. Figures 9c

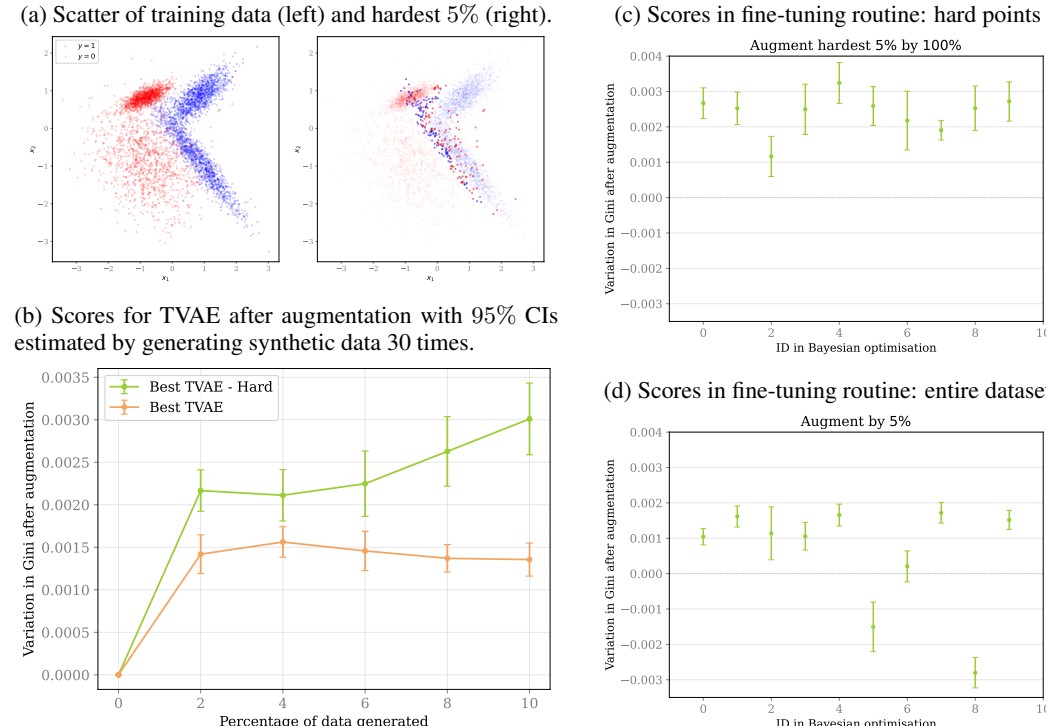

Figure 9: Gini scores under different synthetic data augmentation regimes on the bivariate Gaussian simulated data, with scatterplots of observations and 5% hardest data points.

and 9d display the scores after 10 attempts: the targeted approach results in a larger performance improvement, both for the best attempt and on average across different sets of hyperparameters. Information on the tuning process can be found in Appendix A.3. Finally, we augment by different amounts using the best attempts, with results displayed in Figure 9b: hard points augmentation consistently outperforms the non-targeted approach and improves as more data is added.

## 5 CONCLUSION AND DISCUSSION

In this work, we verified empirically that, when it comes to data augmentation, focusing only on the most challenging points for the model can be beneficial, both in terms of performance improvement and computational efficiency. To achieve this result, we first devised a novel hardness characterizer based on KNN Shapleys, capable of achieving performances comparable to state-of-the-art methods in hardness detection benchmarks on tabular data. Then, we performed a complex fine-tuning routine on synthesizers trained either on the entire dataset or on the hardest points only, allowing to rigorously establish which approach leads to the largest performance improvement.

Methodologically, this work aims at bridging the gap between learning-based hardness characterization and game theoretic data valuation. The benchmarks reported in Section 4.1 represent the first quantitative comparison between a Shapley-based evaluator and existing hardness characterizers. While research on the two topics has often dealt with similar problems, such as mislabeling or data summarization (Paul et al., 2023; Jia et al., 2021), the two paths have never crossed.

Practically, this work attempts to change the perspective on data summarization: while existing literature has mostly focused on pruning the least valuable points from training data (Seedat et al., 2022; Kwon & Zou, 2023), here we propose augmenting the most valuable ones. We demonstrate that not all datapoints are created equal, and some play a more significant role in determining the predictive power of the final model: high-value data dictates which points we should either collect in larger amounts or, when this is not possible, synthetically generate.

REPRODUCIBILITY STATEMENT

The Amex dataset is publicly available on the public data repository Kaggle. This work uses the denoised version by user *raddar*, which is publicly available on Kaggle in `parquet` format. The code to reproduce the results in this work is available in the GitHub repository `anonymised_link`. In particular, code to reproduce the results in Section 4 is available in the `analyses/` folder. The `outputs/` folder contains, in addition to figures and results, the model weights for the best synthesizers. Notice how every method from the `ctgan` and `sdv` packages uses the forked versions (`anonymised_link` and `anonymised_link`) with the custom early stopping. Detailed instructions on how to create a Python virtual environment and install these dependencies are available in the `Makefile`. Details around hyperparameter tuning are also reported in Appendices A.1, A.2 and A.3.

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

# A APPENDIX

## A.1 TRAINING CTGAN AND TVAE – AMEX DATASET

This section provides details about CTGAN and TVAE SDGs on the Amex dataset. A common issue when training GANs is the instability of both generator and discriminator losses. More specifically, the two typically move in opposite ways with oscillatory patterns, making it difficult to decide when to stop training. For this reason, we introduce a novel early stopping condition which tracks epoch after epoch a weighted average of the *Kolgomorov-Smirnov* (KS) statistic between real and synthetic data across all individual features. In particular, we choose to use as weights the feature importances, in order to focus more on the features relevant to XGBoost. The patience is set to 50 epochs with a maximum number of epochs of 500. If the early stopping condition is triggered, the model is

reverted to the best epoch. A large batch size of $10\,000$ is chosen to limit the number of updates per epoch and guarantee a smoother training process. Both the generator and discriminator are trained using the Adam optimizer (Kingma & Ba, 2017), setting the learning rate to $2 \cdot 10^{-4}$, weight decay to $10^{-6}$, and the momentum parameters to $\beta_1 = 0.5$ and $\beta_2 = 0.9$. Figure 10a shows the losses and the statistic epoch by epoch: we can see that the losses move against each other and then more or less converge once the model cannot further improve. We notice oscillatory patterns in the tracked statistic, symptomatic of the training dynamics of the generator and discriminator pair, and the early stopping condition kicking in after around 100 epochs when the weighted KS statistic peaks at $0.83$.

Training of the VAE relies on the Adam optimizer with a learning rate of $10^{-3}$, weight decay of $10^{-5}$, $\beta_1$ of 0.9 and $\beta_2$ of 0.999. To avoid overfitting of training data and the subsequent generation of exact replicas of training points, we stop training once the maximum number of epochs is reached.

The following routine integrating both model selection and hyperparameter tuning is carried out separately for models trained on only the hardest points and models trained using the entire dataset:

- **Model selection**: after training one instance of TVAE and CTGAN to gather initial data, for each candidate model $i$ the *Upper Confidence Bound 1* (UCB1) score is calculated:

$$\text{UCB1}(i) = \bar{\psi}_i + \sqrt{\frac{2 \ln t}{n_i}},$$

    where $\bar{\psi}_i$ is the average score of model $i$, $t$ is the total number of models trained, and $n_i$ is the number of times model $i$ is selected. We select the model with the highest UCB1 score, which results from a balance of exploration (low $n_i$) and exploitation (high $\bar{\psi}_i$).

- **Hyperparameter tuning**: once model $i$ has been proposed, the set of hyperparameters to try is chosen via *Gaussian Process Expected Improvement* (GPEI): a *Gaussian Process* (GP) is fitted to $\{(\mathbf{h}_j, \psi_j)\}_{j=1}^{n_i}$, with $\mathbf{h}_j$ hyperparameters and $\psi_j$ corresponding score, and then the new set to try is chosen to maximize over $\mathbf{h}$ the *Expected Improvement* $\text{EI}(\mathbf{h}) = \mathbb{E}[\max(0, f(\mathbf{h}) - f(\mathbf{h}^*))]$, with $f$ mean function of the fitted GP and $\mathbf{h}^*$ the best set so far.

The score $\psi$ over which the optimization is carried out, is the variation in validation Gini after training XGBoost on the augmented dataset.

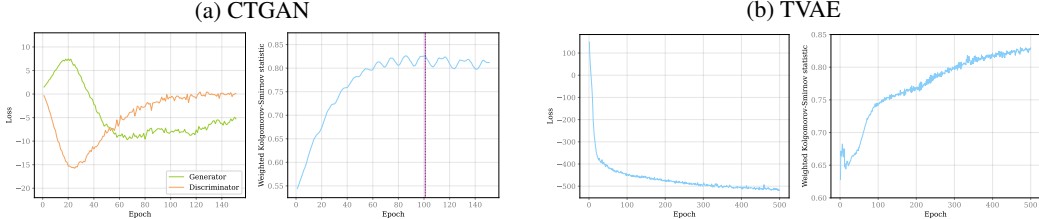

Figure 10: Loss and KS statistics for training CTGAN and TVAE.

## A.2 HYPERPARAMETERS - AMEX DATASET

For TVAE trained on the hardest $10\%$, tuning is performed on the embedding dimension and the architectures of encoder and decoder. We fix the number of hidden layers to two for the encoder and the decoder, tuning the number of units in each hidden layer. Details on the grid search and the best attempt can be found in Table 1, while a parallel coordinates plot is displayed in Figure 11. For CTGAN trained on the hardest $10\%$, tuning is performed on the embedding dimension and the discriminator and generator architectures. We fix the number of hidden layers to two for the discriminator and the generator, tuning the number of units in each hidden layer. Further details can be found in Table 2 and Figure 12. Similar summaries are reported for TVAE (Table 3, Figure 13) and CTGAN (Table 4, Figure 14) on the entire dataset. Notice once again the deterioration in performance with respect to the targeted case.

| Hyperparameter | LB | UB | Default | Best |
|---|---|---|---|---|
| Embedding dim. | 32 | 512 | 64 | **64** |
| Encoder dim. 1 | 32 | 512 | 128 | **128** |
| Encoder dim. 2 | 32 | 512 | 128 | **128** |
| Decoder dim. 1 | 32 | 512 | 128 | **128** |
| Decoder dim. 2 | 32 | 512 | 128 | **128** |

Table 1: TVAE, hard 10%: tuning setup.

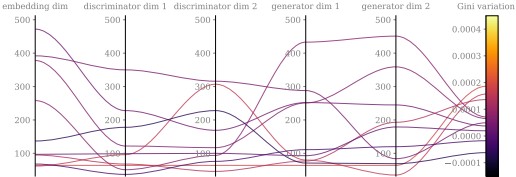

Figure 11: TVAE, hard 10%: parallel coordinates.

| Hyperparameter | LB | UB | Default | Best |
|---|---|---|---|---|
| Embedding dim. | 32 | 512 | 64 | **377** |
| Discrim. dim. 1 | 32 | 512 | 64 | **405** |
| Discrim. dim. 2 | 32 | 512 | 64 | **82** |
| Generator dim. 1 | 32 | 512 | 64 | **171** |
| Generator dim. 2 | 32 | 512 | 64 | **118** |

Table 2: CTGAN, hard 10%: tuning setup.

Figure 12: CTGAN, hard 10%: parallel coordinates.

| Hyperparameter | LB | UB | Default | Best |
|---|---|---|---|---|
| Embedding dim. | 32 | 512 | 64 | **479** |
| Encoder dim. 1 | 32 | 512 | 128 | **249** |
| Encoder dim. 2 | 32 | 512 | 128 | **477** |
| Decoder dim. 1 | 32 | 512 | 128 | **425** |
| Decoder dim. 2 | 32 | 512 | 128 | **33** |

Table 3: TVAE, full data: tuning setup.

Figure 13: TVAE, full data: parallel coordinates.

| Hyperparameter | LB | UB | Default | Best |
|---|---|---|---|---|
| Embedding dim. | 32 | 512 | 64 | **64** |
| Discrim. dim. 1 | 32 | 512 | 64 | **64** |
| Discrim. dim. 2 | 32 | 512 | 64 | **64** |
| Generator dim. 1 | 32 | 512 | 64 | **64** |
| Generator dim. 2 | 32 | 512 | 64 | **64** |

Table 4: CTGAN, full data: tuning setup.

Figure 14: CTGAN, full data: parallel coordinates.

| Hyperparameter | LB | UB | Default | Best |
|---|---|---|---|---|
| Embedding dim. | 16 | 256 | 64 | **78** |
| Encoder dim. 1 | 16 | 256 | 128 | **210** |
| Encoder dim. 2 | 16 | 256 | 128 | **186** |
| Decoder dim. 1 | 16 | 256 | 128 | **107** |
| Decoder dim. 2 | 16 | 256 | 128 | **48** |

Table 5: TVAE, hard 5%: tuning setup.

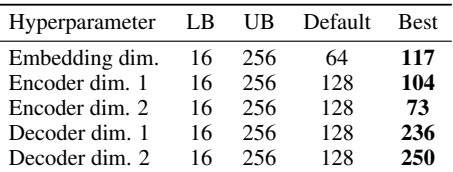

Figure 15: TVAE, hard 5%: parallel coordinates.

| Hyperparameter | LB | UB | Default | Best |
|---|---|---|---|---|
| Embedding dim. | 16 | 256 | 64 | **117** |
| Encoder dim. 1 | 16 | 256 | 128 | **104** |
| Encoder dim. 2 | 16 | 256 | 128 | **73** |
| Decoder dim. 1 | 16 | 256 | 128 | **236** |
| Decoder dim. 2 | 16 | 256 | 128 | **250** |

Table 6: TVAE, full data: tuning setup.

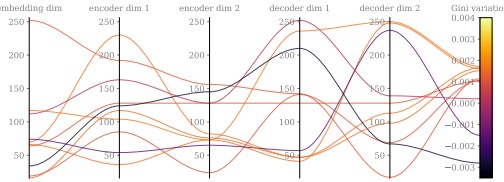

Figure 16: TVAE, full data: parallel coordinates.

### A.3   HYPERPARAMETERS - SIMULATION STUDY

The setup is analogous to Appendix A.2, with the difference that CTGAN has been discarded due to excessive instability during training, and smaller architectures are considered to account for a much simpler dataset. Specifically, Table 5 and Figure 15 report the results for TVAE trained on the hardest 5%, while Table 6 and Figure 16 present TVAE trained on the entire dataset. Once again, we notice worse performance in the non-targeted case in the parallel coordinates plots .

