# OpenReview forum: "Targeted synthetic data generation for tabular data via hardness characterization"
_ICLR.cc/2025/Conference — ICLR 2025 Conference Withdrawn Submission_

### Official Review · Reviewer_AU9t · 2024-10-25

**Soundness:** 2
**Presentation:** 4
**Contribution:** 2
**Rating:** 3
**Confidence:** 5

**Summary:**

The paper proposes targeted synthetic data generation for tabular data using hardness characterization. It introduces a pipeline that focuses on generating synthetic data for the hardest points within the training dataset, as identified by a KNN Shapley-based hardness characterizer. The paper shows that augmenting only the hardest points improves out-of-sample performance and computational efficiency compared to non-targeted data augmentation approaches.

**Strengths:**

Originality: The idea of augmenting the hard samples is novel as is using KNN Shapley values to target hard samples for synthetic data generation is interesting as it differs from the different hardness characterization literature. Although see weaknesses below: neither in KNN shapley novel neither is targeted synthetic data generation.

Quality: The empirical quality is good and the methodology is rigorous and well-executed, with solid empirical benchmarks.

Clarity: The paper is well written and the clear details elucidate the problem and solution well.

Significance: Targeted synthetic data generation could be impactful in improving model robustness, particularly when data is scarce or of poor quality. So overall the problem seems important

**Weaknesses:**

There are three major weaknesses: Novelty, metrics and evaluation

Novelty:
- Use of KNN Shapley: the proposed approach to assess samples using KNN-Shapley isn’t novel, rather already proposed in Jia et al. (https://arxiv.org/abs/1911.07128). So while the application to synthetic data is novel, the proposed KNN-shapley assessment approach option itself is not novel. By itself this isn't an issue. But because the paper proposes to use an off-the-shelf valuation approach, it then warrants assessing why this is the best data valuation approach vs other alternative data valuation approaches (e.g. https://arxiv.org/pdf/2306.10577)

- Targeted synthetic data generation: The paper proposes KNN-shapley to do the targeting. However, the idea of using the targeted samples to condition synthetic generation also mirrors other works from data-centric literature (see https://arxiv.org/abs/2310.16981), which use the hardness ideas mentioned in the paper. It is important to contrast the different approaches in the related work for targeting synthetic data generation vs what's proposed in the paper --- both conceptually and empirically to understand strengths and weaknesses.

Metrics:
- The paper shows the effect of the synthetic data in terms of performance and Gini scores. It is important when using synthetic data to also assess how the statistical fidelity is affected. i.e. is the synthetic data good statistically with respect to real data. It would improve the paper to assess fidelity metrics like KL-Divergence, MMD, Precision and Recall
- Examples of widely used synthetic data metrics (particularly fidelity) can be found in: (1) https://arxiv.org/abs/1806.00035, (2) https://arxiv.org/pdf/2406.13130v1, (3) https://arxiv.org/abs/2102.08921

Evaluation:
The assessment on the Amex dataset is interesting, however to claim generality of the findings a greater diversity of datasets should be explored, either across domains or with different properties (sample sizes, feature dimensionality)

**Questions:**

- Q1: As mentioned in the weaknesses how does the proposed approach to targeted synthetic data generation compare conceptually and empirically to: https://arxiv.org/abs/2310.16981

- Q2: How does the augmentation of only hard samples affect statistical fidelity?

- Q3: Would you say the findings are generalizable to any tabular dataset?

**Details Of Ethics Concerns:**

N..A.

---

> ### Author Response · Authors · 2024-11-20
>
> We thank the reviewer for their valuable comments about our work. Below, we give some comments about the questions they raised:
>
> 1. While the mentioned paper focuses on using the information provided by hardness characterizers to build high quality synthetic datasets, resulting mostly in the conclusion that there is benefit (to statistical fidelity) in pruning ambiguous datapoints (see Data-IQ) and train the generators on easy + hard points only, we aim to show how hardness characterization could guide data augmentation to make it perform better at a lower computational cost. In the setting of the Amex dataset, training synthesizers on easy + hard data would imply training on > 80% of the overall dataset (so roughly the same computational cost) and, with hard points being only 3%, would most likely negate the performance benefit too.
>
> 2. There are two main points to comment on here:
>  - statistical fidelity of the synthetic samples: this is tracked by both visual inspection of the feature distributions and by the weighted average of the Kolmogorov-Smirnov statistic across all features.
>  - how adding synthetic data affects the final dataset: mostly this results in a slight increase in the number of positive labels (hard data, intuitively, tends to belong more to the minority class). As Gini is robust to class imbalances, this does not affect the final performance comparison.
>
> 3. No, the proposed approach only works for classification tasks (binary or multiclass). Also, for datasets of size < 1000, data Shapleys can be calculated exactly thus removing the need for KNN Shapleys as a proxy.

---

### Official Review · Reviewer_sZKp · 2024-10-27

**Soundness:** 1
**Presentation:** 2
**Contribution:** 2
**Rating:** 3
**Confidence:** 3

**Summary:**

This paper presents two main contributions:
* Proposing the use of KNN Shapley values for hardness characterization
* Proposing a synthetic data augmentation pipeline that relies on training a generative model exclusively on hard data points

Both contributions are evaluated empirically. The effectiveness of Shapley values on a benchmark consisting of two real world datasets where it is compared to other hardness characterization baselines.
The synthetic data augmentation pipeline is applied to a large credit default prediction dataset, demonstrating modest performance gains, with a slight advantage observed when augmenting using only hard examples.

**Strengths:**

The paper is generally well-written, with a coherent presentation of the main ideas.

The authors go to great lengths to justify every decision and significant care seems to have been taken to achieve a sensible evaluation setup that appropriately accounts for the inherent variance in the methods.

**Weaknesses:**

**Main Points**

The main weaknesses of the paper lie with the experimental setup:
1. The main claims of the paper on data augmentation are supported by experiments on a single real-world dataset. This seems insufficient to draw meaningful conclusions about the wider applicability of the method. I would be interested to know if the conclusions hold on a wider range of datasets with different sizes and tasks.
2. Likewise, the results on KNN Shapleys as a hardness characterization are supported by evaluations on a benchmark of only 2 tabular datasets.
3. It would also be interesting if KNN Shapleys were compared to other hardness measures on the downstream synthetic data augmentation task.
4. More recent methods such as [1], [2] or [3] have shown significant improvements for synthetic data generation in tabular data over TVAE and CTGAN. It would be more relevant to evaluate the synthetic data generation pipeline with these methods.



**Minor Point**

The authors might want to be more precise with statements that can be easily misinterpreted such as (page 8):
> For models trained on the 10% hardest points we augment by 100%, while for models trained using
> the entire dataset we augment by 10%, thus guaranteeing the same amount of synthetic samples
> across all experiments.

I am assuming the 100% refers to the size of the training set used for the generative model (10% of the original training set).

**References**

[1] Akim Kotelnikov, et al. TabDDPM: Modelling Tabular Data with Diffusion Models, 2022. https://arxiv.org/abs/2209.15421

[2] Jayoung Kim, et al. STaSy: Score-based Tabular data Synthesis, 2022. https://arxiv.org/abs/2210.04018

[3] Hengrui Zhang, et al. Mixed-Type Tabular Data Synthesis with Score-based Diffusion in Latent Space, 2023. https://arxiv.org/abs/2310.09656

**Questions:**

1. The authors set aside a significant portion of data for hardness characterization. This does not seem ideal in a scenario where data augmentation is required since the additional data could be used for training the model instead. Do other hardness measures also require such hold-out sets?
2. Is there any downside to adding back the held out data together with the generated synthetic data when retraining the model? If not why wasn't this part of the experimental setup?
3. KNN assumes a distance metric over the input space. This is problematic for structured tabular data with mixed categorical and numerical features where a natural distance metric is not available and each feature can have different importances and scales. How was the distance chosen for KNN and how can the authors justify it?

---

> ### Author Response · Authors · 2024-11-20
>
> We thank the reviewer for their valuable comments about our work. Below, we give some comments about the questions they raised:
>
> 1. The necessity for a hold-out dataset is a weakness of all Shapley-based data valuators. Data-IQ, for example, does not require a hold-out set. At least on the Amex dataset, we showed that 10% augmentation of hard data outperforms adding back the 10% we used as a hold-out set, meaning there is still a net gain.
>
> 2. We do not see any issues in adding back the hold-out set together with synthetic data. The focus of the work was more on proving that there is value in identifying the most valuable training samples before data augmentation, but we agree that if the objective is to get the largest possible performance improvement, then the held-out set should be added back.
>
> 3. The data was standardised to zero mean and unit variance before calculating KNN Shapleys using the Euclidean distance. While using different distances for numeric and categorical features may be beneficial, the bottleneck here is mainly the computational cost.

---

### Official Review · Reviewer_XpsT · 2024-10-31

**Soundness:** 2
**Presentation:** 3
**Contribution:** 2
**Rating:** 3
**Confidence:** 2

**Summary:**

This paper proposes to (1) use data valuation methods rooted in game theory to characterize samples that are difficult / hard to learn, and (2) improve model learning / performance by generating synthetic data around these hard / highly valued samples. It is an empirical study that extends our understanding of known methods (KNN Shapley, Data-IQ, SMOTE, CTGAN, TVAE).

**Strengths:**

This work proposes to change the perspective on data distillation, i.e. to augment the most valuable points in training data, instead of pruning the least valuable ones. The idea itself is original and valuable, but...

**Weaknesses:**

... my impression is that this work resembles a shorter workshop paper rather than a complete contribution, i.e.:

0. (For context, there are no theoretical contributions in the paper, which is not a weakness in isolation.)
1. **Empirical evaluation.** There is little signal that the proposed method works. Although I am generally in favor of interesting, simple ideas, even insightful negative results, experiments are lacking:
  - A) Overall results with only 4 tabular dataset-model pairs are rather anecdotal and insufficient to make meaningful conclusions. What holds back from doing a comprehensive benchmark on 10++ datasets/models (e.g. OpenML-CC18, OpenML-CTR23 benchmarks) that would empirically show the method's superiority?
  - B) Results presented in Sec. 4.1 are unconvincing, i.e. the method performs well on one of two datasets. The heatmap is unclear (I can't see differences between the columns/rows); why not present AUPRC in a table / lineplot / barplot, also with deviations / confidence? Moreover, it seems the mentioned conclusion about differences between the methods (L297-L302) could be derived without doing this benchmark at all.
  - C) Regarding Sec. 4.2, L468-473: I can instead argue that the chosen dataset + model (Amex + XGBoost) is not a good example to show the added value of the proposed method. Mind you, this improvement by 0.0004 seems to be the paper's main result.
  - D) The last Section 4.3 with Figure 9 is an ablation "to verify the robustness of the results" on synthetic data instead of making a convincing case on more various datasets.
2. **Computational efficiency.** There are multiple unsupported claims made about the method's efficiency, also as compared to baselines. There are no experimental results to support these claims, e.g. analyzing the computational speed of KNN Shapley, showing its impact on the XGBoost model's convergence and computational time.
3. **Redundancy in presentation?** Subjectively, the paper has overlong descriptions and too many figures, if only to fill the 10 pages. There are 9 figures containing 26 subfigures / tables, potentially obfuscating the unseen key result / contribution. See details below.

**Questions:**

0. **Redundancy?** Only for example:
  - Section 2. "Background" spans 2 pages describing in (too) much detail the already published methods (KNN Shapley, Data-IQ, SMOTE, CTGAN, TVAE), which are later applied but not modified or analyzed.
  - L315-L348 are details about training a single XGBoost model that would usually be found in the Appendix.
  - L351-369 discusses results only for the Data-IQ method (Seedat et al., 2022). Virtually Section 4.2.1 reads like a critique of Data-IQ (Seedat et al., 2022) instead of showing the applicability of the proposed method. Perhaps it can be better communicated to the reader at the beginning of this section / in its title (currently "Harndess characterization").
  - Figure 7 provides no valuable information.
1. The font size in Figures 1, 6, 7, 8(a,b), and 9(a) is too small. This could be fixed by saving the figures to smaller sizes so that all the elements will increase in comparison.
2. Please clarify: how do you label generated samples?
3. Clarify in the paper: In Sec. 3, what is model A? Does it need to be a KNN? (no)
4. Discuss in the paper: Does this method work for tabular regression tasks? (no)
5. In L300, we read "not requiring model training", but then L376 says "To choose the best K, validation Gini is monitored as we re-train XGBoost after gradually removing the hardest datapoints".
6. What is the motivation behind using the normalized Gini coefficient "2*AUCROC − 1" instead of F1?

There is a related work that came to mind when reading the paper:

- How does this relate to the concepts of dataset distillation (arXiv:1811.10959) and dataset condensation (ICLR 2021)?
- Can alternative data generators, e.g. TabDDPM (ICML 2023) or Adversarial random forests (AISTATS 2023), be used in this framework?

---

> ### Author Response · Authors · 2024-11-20
>
> We thank the reviewer for their valuable comments about our work. And many thanks for the valuable feedback on the presentation! Below, we give some comments about the questions that were raised:
>
> 2. The label column is treated as any other column by the synthetic data generator. It knows from the manually specified schema it is binary (so it only generates 0 or 1) and generates values according to the model employed (CTGAN or TVAE).
>
> 3. We agree: this section should have been made clearer. As the formula for KNN Shapleys is completely model agnostic (it effectively only considers the distance between training and test data), model A can be any model.
>
> 4. No. Unfortunately the field of data valuation for regression tasks seems to be in general significantly less mature than the one for binary or multiclass classification.
>
> 5. With "not requiring model training" we refer to the model agnostic nature of KNN Shapleys calculation, whereas with "to choose the best K, validation Gini is monitored as we re-train XGBoost after gradually removing the hardest datapoints" we refer to the strategy used to choose the optimal K: we plot test set performance as we gradually remove training data (low KNN Shapleys first) and consider as best the K for which we see the fastest drop. This iterative procedure is what requires XGBoost re-training.
>
> 6. It is important to use a metric which is insensitive to class imbalance (like AUCROC or Gini) as data augmentation of hard data will inevitably change the ratio of positive/negative labels in the final training dataset. As for why we chose Gini over AUCROC, this is just more common in the credit setting (and it was used in the original Kaggle competition on the Amex dataset).

---

### Official Review · Reviewer_bH8o · 2024-11-12

**Soundness:** 3
**Presentation:** 3
**Contribution:** 3
**Rating:** 6
**Confidence:** 3

**Summary:**

The paper proposes a novel synthetic data augmentation approach focused on enhancing binary classification models, particularly on tabular data. It introduces a targeted synthetic data generation pipeline, using KNN-based Shapley values to identify the "hardest" points in the training data. By generating synthetic data specifically for these challenging data points, the paper aims to strengthen the model’s ability to generalize to new, unseen data while reducing computational overhead compared to non-targeted augmentation methods. Empirical evaluations demonstrate the benefits of this approach in improving out-of-sample prediction quality, particularly in applications like credit default prediction.

**Strengths:**

1. Originality: The paper introduces a novel approach to synthetic data generation, focusing on "hard" data points identified via KNN-based Shapley values. This method augments only challenging points, enhancing model robustness and bridging data valuation with data augmentation.

2. Quality: Extensive experiments on real and simulated datasets. Detailed benchmarking against existing hardness characterizers and comprehensive hyperparameter tuning validate the method, particularly w.r.t. its computational efficiency and model robustness.

3. Clarity: The paper is well-structured and clear. The figures and precise mathematical formulations make it easier to understand the paper.

4. Significance: This approach has broad applicability for fields like finance and healthcare, where robust tabular data predictions are essential. By focusing on challenging data points, the method not only improves model performance but also offers a cost-effective solution.

In summary, the paper makes a strong contribution to synthetic data generation and introduces novel methods and demonstrates practical benefits for high-stakes applications involving tabular data.

**Weaknesses:**

1. The paper's focus is restricted to binary classification tasks, which limits the generalizability of the approach. Could the authors consider experimenting with multi-class tasks at least?
2. The paper relies heavily on KNN Shapley values for hardness characterization. It might be interesting to explore other methods such as gradient-based techniques (e.g., GraNd) or dropout-based uncertainty measures. Would the authors be open to trying this?
3. The method relies on specific synthetic data generators like CTGAN and TVAE, which may not perform as effectively on all types of tabular data (e.g., highly skewed or sparse datasets).
4. The study uses a fixed threshold (e.g., augmenting 10% of the hardest points) which seems a bit hard to justify. Could the authors consider other alternatives?

**Questions:**

See above in weaknesses for suggestions/questions.

---

> ### Author Response · Authors · 2024-11-20
>
> We thank the reviewer for their valuable comments about our work. Below, we give some comments about the questions they raised:
> 1. Both Data-IQ and KNN Shapleys can easily be extended to multiclass classification, and the same holds true for CTGAN and TVAE. This means that experiments on multi-class tasks are possible. In future versions of the manuscript, we will attempt to provide additional examples on this.
> 2. There are several possible approaches to data valuation and hardness characterization. Unfortunately, all Shapley-based data valuators are just too computationally expensive for datasets of the size of the Amex dataset, and most hardness characterizers (like GraNd and EL2N) are designed for neural networks and cannot be trivially adapted for XGBoost. Given the focus of this work on gradient boosting on large tabular datasets (very common in industry), KNN Shapleys and Data-IQ were the best options available. In future versions of this work, we will further emphasise that the choice of KNN Shapleys and Data-IQ is for computational reasons on large datasets.
> 3. There are surely more advanced synthetic data generators than CTGAN or TVAE. We focused on these as they are available in most commonly used Python packages (SDV, synthcity), and because the focus of the work was rather on the idea of combining hardness characterization combined with a data augmentation pipeline than on the employment of the most state-of-the-art generators.
> 4. We agree: the fixed threshold is difficult to justify. As experimenting with different thresholds is computationally expensive (considering the hyperparameter tuning performed on the generators), the idea is to plot the test set performance as we remove the most valuable training data and look for an “elbow” where the performance deterioration slows down. That is how we chose the 10%.

---

### Author Response · Authors · 2024-11-20

We would like to sincerely thank the reviewers for their thoughtful and constructive feedback on our paper. The issues raised are insightful and highlight areas requiring significant improvement and refinement. After careful consideration, we have concluded that addressing these issues thoroughly and effectively would necessitate more time than the two weeks provided for the rebuttal period. Therefore, we have decided to withdraw our paper from further consideration at this time. We deeply value the reviewers' comments and will carefully incorporate them as we work on a future, and hopefully improved, version of this work. We provide detailed responses to address the comments of each reviewer.

---

### Note · Authors · 2024-11-20

**Comment:**

We would like to sincerely thank the reviewers for their thoughtful and constructive feedback on our paper. The issues raised are insightful and highlight areas requiring significant improvement and refinement. After careful consideration, we have concluded that addressing these issues thoroughly and effectively would necessitate more time than the two weeks provided for the rebuttal period. Therefore, we have decided to withdraw our paper from further consideration at this time.

**Withdrawal Confirmation:**

I have read and agree with the venue's withdrawal policy on behalf of myself and my co-authors.